# Proteomics Analysis of *E. angustifolia* Seedlings Inoculated with Arbuscular Mycorrhizal Fungi under Salt Stress

**DOI:** 10.3390/ijms20030788

**Published:** 2019-02-12

**Authors:** Tingting Jia, Jian Wang, Wei Chang, Xiaoxu Fan, Xin Sui, Fuqiang Song

**Affiliations:** 1Engineering Research Center of Agricultural Microbiology Technology, Ministry of Education, Heilongjiang University, Harbin 150500, China; 18945081056@163.com (T.J.); 13796672755@163.com (J.W.); changwei77@126.com (W.C.); fan_xiao_xu@126.com (X.F.); xinsui_cool@126.com (X.S.); 2Heilongjiang Provincial Key Laboratory of Ecological Restoration and Resource Utilization for Cold Region, School of Life Sciences, Heilongjiang University, Harbin 150080, China

**Keywords:** arbuscular mycorrhizal fungi, salt stress, *E. angustifolia*, proteomics

## Abstract

To reveal the mechanism of salinity stress alleviation by arbuscular mycorrhizal fungi (AMF), we investigated the growth parameter, soluble sugar, soluble protein, and protein abundance pattern of *E. angustifolia* seedlings that were cultured under salinity stress (300 mmol/L NaCl) and inoculated by *Rhizophagus irregularis* (RI). Furthermore, a label-free quantitative proteomics approach was used to reveal the stress-responsive proteins in the leaves of *E. angustifolia*. The result indicates that the abundance of 75 proteins in the leaves was significantly influenced when *E. angustifolia* was inoculated with AMF, which were mainly involved in the metabolism, signal transduction, and reactive oxygen species (ROS) scavenging. Furthermore, we identified chorismate mutase, elongation factor mitochondrial, peptidyl-prolyl cis-trans isomerase, calcium-dependent kinase, glutathione S-transferase, glutathione peroxidase, NADH dehydrogenase, alkaline neutral invertase, peroxidase, and other proteins closely related to the salt tolerance process. The proteomic results indicated that *E. angustifolia* seedlings inoculated with AMF increased the secondary metabolism level of phenylpropane metabolism, enhanced the signal transduction of Ca^2+^ and ROS scavenging ability, promoted the biosynthesis of protein, accelerated the protein folding, and inhibited the degradation of protein under salt stress. Moreover, AMF enhanced the synthesis of ATP and provided sufficient energy for plant cell activity. This study implied that symbiosis of halophytes and AMF has potential as an application for the improvement of saline-alkali soils.

## 1. Introduction

Salt stress is one of the most important abiotic stresses and limiting factors for plant growth and agricultural production. It is a major abiotic stress in the world. Land salinization causes many ecological and environmental problems, such as soil erosion, land desertification, forest and grassland degradation, and biodiversity reduction [1]. At present, along with the increase in soil salinization and secondary salinization, it is estimated that 30% of the arable land in the world will disappear in the next 25 years, and 50% by the middle of the 21st century [2,3]. Hence, the question of how to treat saline alkali soil has attracted widespread attention around the world. In recent years, it was demonstrated that using biological means to treat soil salinization is highly efficient, and environmental and sustainable, thus providing a new breakthrough method for saline alkali land treatment.

*E. angustifolia*, a member of the family, Elaeagaceae, is a deciduous tree that is widespread in the vast desert and semidesert in the Northwest of China. A few varieties of the species, *E. angustifolia*, can survive in the Gobi, such as the desert and saline, and is called the “treasure tree” locally. It is important to further improve the salt tolerance of *E. angustifolia* using biotechnology under saline-alkali conditions. Arbuscular mycorrhizal fungi (AMF) exist widely in soil and form a mutualism system with most higher plants [4,5]. Plants are subjected to salt stress in the presence of a high salinity in the soil, which reduces the absorption and transport of water, inhibits the metabolic process, and affects nutrient absorption and the cell infiltration balance, resulting in the fragmentation of the horny layer of plants and leakage of the cell membrane. This leads to plant growth retardation. AMF can adapt to the saline soil habitat and survive in a heavy salt environment, indicating that AMF is adaptable to saline soil [6]. Previous studies have shown that symbiosis between AMF and plants under salt stress can promote plant growth and improve plant salt tolerance [7,8]. Therefore, the symbiosis of halophytes with AMF has great potential for the improvement of salt resistance and restoration of saline-alkali land, which has been a major research field globally.

Previous studies on the application of proteomics technology revealed the salt tolerance of plant leaves [9,10,11,12,13,14,15]. However, the response mechanism of mycorrhizal plants to salt stress needs to be further revealed. In this study, the symbiosis of AMF *Rhizophagus irregularis* (RI) and the salt-tolerant plant, *E. angustifolia*, was used as a breakthrough point. The stress-responsive proteomics in the leaves of *E. angustifolia* were detected under salt stress conditions. These results will provide more information for the understanding of the function of AMF in the improvement of plant salt tolerance.

## 2. Results

### 2.1. Growth of E. angustifolia under Salt Stress and Colonization of AMF in the Plant Roots

As shown in Figure 1A, both mycorrhizal and non-mycorrhizal seedlings grew well in the treatments lacking salt, but the mycorrhizal seedlings’ leaves grew stronger than the non-mycorrhizal seedlings; some leaves of the mycorrhizal and non-mycorrhizal seedlings were yellow during salt stress, however, the number of withered leaves of the mycorrhizal plants was significantly less than that of the non-mycorrhizal plants.

The typical AMF morphological structure was detected in inoculated *E. angustifolia* roots, including vesicles and arbuscules (Figure 1B). The maximum AMF colonization percentage of the root reached more than 90% at approximately 100 at approximately 30 days after inoculation. The maximum AM colonization percentage of the root reached more than 90% at approximately 30 days after salt stress. At the same time, no colonization was found in the non-inoculated seedlings. This result shows that *E. angustifolia* and *R. irregularis* established a vigorous symbiosis. 

### 2.2. Effects of RI and CK on Height, Diameter, and Roots of E. angustifolia under Salt Stress

Salt stress decreased the height, diameter, length, and area, but mycorrhizal seedlings had a greater height, diameter, length, and area than non-mycorrhizal seedlings during salt stress (Table 1). During the 300 mmol/L NaCl treatment, the height, diameter, length, and area of the mycorrhizal seedlings increased by 9.1%, 20.8%, 17.4%, and 35.5%, respectively, compared with those of the non- mycorrhizal seedlings. AMF inoculation significantly enhanced the growth parameter of *E. angustifolia* seedlings in the presence of 300 mmol/L NaCl.

### 2.3. Effects of RI and CK on the Soluble Sugar Content, Soluble Protein Content in the Leaves of E. angustifolia under Salt Stress

As shown in Figure 2A, salinity stress caused a significant decline in the leaf soluble protein content of mycorrhizal and non-mycorrhizal seedlings, while mycorrhizal seedlings had a higher leaf soluble protein content than non-mycorrhizal seedlings during salt stress treatments. As shown in Figure 2B, AMF inoculation significantly promoted the leaf soluble sugar content in the treatments lacking salt. The soluble sugar content in the leaves of mycorrhizal and non-mycorrhizal seedlings increased, but mycorrhizal seedlings had a higher leaf soluble sugar content than that of the non-mycorrhizal seedlings during salt stress. 

### 2.4. Effect of RI on Protein Abundance under Salt Stress

In the CK, AM-NaCl, AM, and AM-NaCl groups, a total of 25,082 peptides and 4349 proteins were identified in the *E. angustifolia* seedlings. The number of proteins identified in the three replicates of each treatment group is shown in Figure 3. Quantifiable proteins were identified in at least two of the three replicates for further analysis. The significance of the differential proteins’ abundances was filtered by the ratio > ±2 and *p* value < 0.05. The numbers of the differentially abundant proteins between treatments (NaCl vs CK, AM vs CK, AM + NaCl vs AM, and AM + NaCl vs NaCl) are shown in Table 2.

### 2.5. Functional Classification of Proteins

#### 2.5.1. Salt Tolerance-Related Proteins Induced by Symbiosis

As shown in Table 2, a total of 187 differentially expressed proteins were identified in the AM vs the CK group. The 186 proteins were compared with the AM + NaCl vs the AM group; a total of 112 were found in the AM + NaCl vs the AM group. Among the 112 proteins, four proteins were highly abundant, 25 were newly expressed proteins under salt stress, and 14 proteins were identified as symbiotic salt tolerance related proteins after referring to many academic documents, as shown in Table A1. These 14 proteins are beneficial to the maintenance of AMF-*E. angustifolia* symbiosis and improved the salt tolerance of the plant under salt stress.

#### 2.5.2. Functional Classification of Salt Tolerance-Related Proteins Induced by Symbiosis

Blast2GO (Version 3.3.5) was used to annotate the biological functions of the targeted proteins. These proteins were divided into seven groups (Figure 4), including metabolism, signal transduction, redox, transport, cytoskeleton, protein synthesis, protein folding, and degradation (Table A1). Among them, the proportion of metabolic and protein folding related proteins were the largest, which was 22%. The second category was related to redox, transport, and cytoskeleton, which was 14%. The third category included signal transduction-related and protein synthesis proteins.

#### 2.5.3. Salt-Induced Mycorrhizal Protein

As shown in Figure 5, after a comparison of differentially expressed proteins between NaCl vs CK and AM + NaCl vs AM groups by VENNY 2.1 (http://bioinfogp.cnb.csic.es/tools/venny/index.html), 121 out of 392 proteins in the AM + NaCl vs AM group were identical with those in the NaCl vs CK group. It is suggested that these 121 proteins are salt-tolerant related proteins of *E. angustifolia* under salt stress, and are not caused by mycorrhizal. However, 271 proteins not in the NaCl vs CK group were considered to be salt-induced mycorrhizal proteins. The mycorrhizal proteins produced by AMF-*E. angustifolia* symbiosis to adapt to salt stress under salt stimulation. Thus, the salt tolerance of plants can be improved.

#### 2.5.4. Functional Classification of Salt-Induced Mycorrhizal Protein

A total of 57 out of 271 proteins were previously reported to be key proteins for the salt response, as shown in Table A2. These were divided into 10 groups by Blast 2 GO analysis (Figure 6). In these functional groups, the first class (23%) are proteins related to metabolism. There are nine (accounting for 14%) different expressed proteins in the signal transduction pathway, which is the second class. Meanwhile, the other 10 functional groups are also involved in protein redox, protein synthesis, photosynthesis, energy, transport, the cytoskeleton, and stress response related proteins.

## 3. Discussion

AMF and salt affected the obvious physical appearance of *E. angustifolia* leaf, and both have a clear interaction. For this reason, we think that AMF has a great influence on salt habitats and that, moreover, salt stress is also a factor in this influence. On the one hand, compared with the AM vs CK group and the AM + NaCl vs AM group, the salt-tolerant proteins caused by the symbiosis could be identified; on the other hand, by comparing the differentially abundant proteins of the NaCl vs CK group and the AM + NaCl vs AM group, the protein for the symbiosis response caused by salt treatment could be identified. Therefore, these two ways were selected to discuss how symbiosis responds to salt stress at the protein level (Figure 7).

### 3.1. Proteins Related to Metabolism

Metabolism, consisting of basic physiological processes, maintains a series of activities of a living organism. This research shows the most important factor in the leaves of the *E. angustifolia* after salt stress was the abundance of secondary metabolism-related proteins. In secondary metabolism, especially the metabolism of the protein, phenylpropane and flavonoids increased significantly under salt stress.

Studies have shown that secondary metabolites of plants change during the symbiosis of mycorrhizal fungi and plants [16]. These secondary metabolites play an important role in the symbiotic relationship between plants and mycorrhizal fungi [17]. For example, the content of lignin and soluble phenol in the tomato was increased by inoculation with Arbuscular mycorrhizal fungi. Flavonoids can promote spore germination and mycelium growth and increase the content of flavonoids after mycorrhizal formation [18,19]. In this study, there were two pathways involved in metabolism-related proteins, as shown in Figure 8.
In this study, we found a chorismate mutase (CM), which catalyzes the conversion of branched acid to prebenzoic acid. Prebenzoic acid can produce phenolic compounds through the phenylpropane metabolic pathway, including phenylalanine (Phe), tyrosine (Tyr), anthocyanin, and tannin [20,21]. Phenylpropane metabolism is indirectly generated by the shikimic acid pathway. This pathway might play an important role in plant stress defense. We found three proteins that relate to the phenylalanine metabolic pathways, including shikimate O-hydroxycinnamoyltransferase, cinnamyl alcohol dehydrogenase, and caffeoyl-CoA O-methyltransferase. Flavonoids are synthesized by the condensation of phenylpropane derivatives with malonate monoacyl coenzyme A. In addition, shikimic acid O-hydroxyacinnamate transferase and caffeoyl coenzyme A-O-methyltransferase were also involved.In this study, we found that the phosphoribosyltransferase (APT) was up-regulated, which was the first key enzyme in the tryptophan production reaction of o-aminobenzoic acid. Phosphorylribosyltransferase activity of o-aminobenzoic acid was enhanced, which accelerated the synthesis of tryptophan in plants under salt stress. It is well known that tryptophan is a precursor of auxin (indole acetic acid) as well as protein synthesis in plants. Auxin response was also identified in the leaves of *E. angustifolia*. We deduced that these two pathways synthesize auxin to maintain the growth and metabolism of mycorrhizal plants under salt stress.


Alkaline neutral invertase is involved in the decomposition of sucrose into glucose and fructose and plays an important role in plant growth and development. The study showed that NaCl and PEG (Polyethylene glycol) stress increased the differential expression of sugar cane *SoNIN1* in the root and leaf [22], and the alkaline neutral transformation enzyme was involved in the stress response. This study found that after inoculation of AMF to the *E. angustifolia* leaf in the treatment of salt stress, alkaline neutral invertase increased its expression, thereby improving the soluble sugar content of the *E. angustifolia* leaf, and providing more sugar for plant metabolism.

### 3.2. Protein Synthesis, Folding, and Degradation Related Proteins

Protein synthesis plays an important role in plant growth under abiotic stress. We found two proteins (ubiquitin-60S ribosomal L4 and 60S acidic ribosomal P1) were up-regulated in the mycorrhizal plant. Similar ribosomal proteins were also found in studies [23]. One study showed that mitochondrial elongation factors can extend peptide chains more [24], and this was also found to be up-regulated in this study. These proteins related to protein synthesis increase the tolerance of the mycorrhizal plant to salt stress by increasing the expression level.

Proteins can lose their biological functions due to denaturation under adverse conditions. Correct folding and degradation of proteins are key to the maintenance of normal cell functions. Molecular chaperones and folding enzymes play an important role in the maintenance of the natural conformation of proteins, which can help them fold correctly [23,25]. In this study, we found that four folding enzymes, peptide-based prolyl cis-trans isomerases, and four molecular chaperones were up-regulated under salt stress, including peptidyl-prolyl cis-trans isomerase FKBP12, FKBP-type peptidyl-prolyl cis-trans isomerase 5 isoform 1, peptidyl-prolyl cis-trans isomerase CYP18-1, peptidyl-prolyl cis-trans isomerase FKBP62, prefoldin subunit 1, prefoldin subunit 2, heat shock 70 kDa partial, and small heat shock protein 17.3 kDa. Therefore, the up-regulation of these four peptidyl prolyl cis-trans isomerases and four molecular chaperones completes the correct folding of proteins and helps the mycorrhizal *E. angustifolia* resist salt stress. 

E3 ubiquitin ligase (UPL3) and ubiquitin-like 1-activating enzyme (E1 B) increased their expression under salt stress. Studies have shown that E3 ubiquitin protein ligase, ubiquitin activation enzyme E1, and ubiquitin ligase all catalyze ubiquitin to their target proteins. Moreover, the specificity of the ubiquitin pathway is controlled by E3, which is pertinent because it can provide the greatest response to environmental stress by regulating the transcription factor of the downstream stress response [26].

### 3.3. Signal Transduction-Related Proteins

Plants exposed to an adverse environment for a long time will produce a complex system of signal sensing and transduction. It is particularly important to understand how mycorrhizal plants perceive and transmit this stress signal, as well as method of improving the plant salt tolerance under salt stress.

In this study, a series of proteins related to signal transduction were screened out, including G protein, phospholipase C, plasma membrane Ca^2+^ transporter ATPase (PMCA), calcium-dependent protein kinase (CDPKs), calmodulin (CaM), and calcium binding (CML), and were up-regulated. They communicated with each other, thus completing the process of perceiving and transmitting stress signals.

G protein, also known as signal-converting protein or coupling protein, can specifically bind and recognize signals on the cell membrane, and produce intracellular signals with the medial membrane effector enzyme (phospholipase C), which plays a role in signal transduction. After transmembrane conversion, extracellular signals are further transmitted and expanded through Ca^2+^ (second messenger) signals, which eventually lead to a series of physiological and biochemical reactions in cells. Ca^2+^ participates in metabolic pathways that are mainly dependent on changes in the Ca^2+^ concentration [27], and this process is achieved by the various calcium transport systems distributed in the cell organelles and cell membranes [28]. The changes in the Ca^2+^ concentration in the various processes in this study depended on the plasma membrane Ca^2+^ transporter, ATPase (PMCA), which is the main Ca^2+^ transporter, transporting Ca^2+^ to the extracellular space at the cost of hydrolyzing an ATP molecule. Studies have shown that the calcium-dependent protein kinase (CDPKs), calmodulin, calmodulin, and the calcium phosphatase B protein, are involved in cell signal transduction and responses to specific stimuli [29,30]. In this study, the Ca^2+^ concentrations maintained a steady state in the cell wall, mitochondria, endoplasmic reticulum, and vacuole, while the concentration in the cytoplasm was low. After salt stimulation, the Ca^2+^ concentration in the cytoplasm increased significantly. On the one hand, Ca^2+^ directly binds to the calmodulin or the calmodulin, transfers the received signal to the protein kinase and stimulates its activity, or the activity of calcium depends on the protein kinases (CDPKs), which are directly stimulated by the Ca^2+^ signal, and directly participate in and cause subsequent physiological responses through these two processes. These two processes directly participate in and cause subsequent physiological responses.

### 3.4. ROS Scavenging-Related Proteins

Active oxygen species (ROS) are usually accumulated in plants under salt stress [31], which can be used as signal molecules to activate the plant stress defense system [32]. However, all ROS are very harmful to organisms at high concentrations, leading to cell membrane peroxidation, destruction of enzyme activity, and eventually leading to cell inactivation. Therefore, the removal of ROS can resist salt damage and improve the salt tolerance of plants. In this study, thioredoxin (TRX) and glutathione (GRX) were involved in ROS scavenging as redox enzymes. In these plants, based on the TRX redox system that runs in various kinds of organelles, including the cytoplasm, mitochondria, and chloroplasts [33,34,35,36], it was shown that TRX plays a key role in plant redox regulation. At present, many GRXs have been identified in different plants. For example, over-expression of the tomato *SLGRX1* gene can enhance the plant’s resistance to oxidative stress, drought resistance, and salt pollution in *Arabidopsis*. In contrast, silencing this gene leads to an increased insensitivity to stress in the tomato [37]. Meanwhile, the gene silencing increases the membrane lipid peroxide level and the accumulation of ROS and suppresses the activity of antioxidant enzymes under high-temperature stress. This suggests that GRXs are involved in the regulation of the redox status and the response to high-temperature stress.

In our study, three glutathione s-transferase (GST) and two glutathione peroxidase (GPX) were found. Studies have reported that to prevent ROS damage, the amount of glutathione transferase (GST) in plants will increase, which can catalyze the removal of ROS in plants [38]. Glutathione peroxidase (GPX) is a sulfur-containing peroxidase, which can remove hydrogen peroxide, organic hydroperoxides, and lipid peroxides from the organism, and block further damage of ROS to the organism [39,40]. It has been shown that chloroplast glutathione transferase plays a very important role in the resistance of low concentrations in Stargrass seedlings [41]. When plants are exposed to high salt stress, the expression activity of GPX will be enhanced and the tolerance of plants to salt stress will be enhanced [42,43,44]. In addition, excessive expression of GST/GPX in transgenic tobacco under salt stress conditions for seed germination and seedling growth were improved more than for the control group, suggesting that GST/GPX increases the ROS removal in plants, protects plants from oxidative damage, and maintains the growth of plants [45,46]. These results suggest that the GST/GPX system is a key factor in the improvement of the salt tolerance of plants through its ROS scavenging ability under salt stress. Peroxidase (POD) is one of the key enzymes in plants under stress conditions in the enzymatic defense system. It cooperates with superoxide dismutase and catalase to remove excess ROS to improve plant resistance.

“Arbuscular Mycorrhizal Symbiosis Modulates Antioxidant Response and Ion Distribution in Salt-Stressed *Elaeagnus angustifolia* Seedlings”- Changes of SOD, CAT, POD, and APX activities in the leaves of *E. angustifolia* inoculated with AMF and those of non-inoculated *E. angustifolia* under 0 and 300 salt concentrations were analyzed [47].

As shown in Figure 9, mycorrhizal seedlings had a higher leaf SOD (superoxide dismutase), CAT (Catalase), POD (Peroxidase), and APX (ascorbate peroxidase) activity than that of the non-mycorrhizal seedlings during salt stress. The mycorrhization of the plants led to increased levels of leaf antioxidant defense systems during stress conditions. The results obtained at the protein level are consistent with the results of the apparent physiological indicators in this study. Therefore, we can conclude that TRX/GRX, GST/GPX, and POD are up-regulated in mycorrhizal plants under salt stress, which results in a significant increase of SOD, CAT, POD, and APX activities in plant leaves, thus improving the salt tolerance of mycorrhizal plants.

### 3.5. Energy-Related Protein

Mitochondria are the main site of oxidative phosphorylation and the synthesis of adenosine triphosphate (ATP) in cells, and they provide energy for cell activity. NADH dehydrogenase, cytochrome C oxidase, iron sulfur protein NADH dehydrogenase, and ATP synthase up-regulation were found in this study as complex compounds I, IV, and V participated in the mitochondrial electron transport. The mitochondrial electron transport chain is also called the respiratory chain, and the cells transfer electrons obtained from the oxidation of macromolecules via I, II, III, and IV complexes and the energy produced by the electron transfer maintains the proton gradient of the mitochondrial inner membrane, which is utilized by complex V (ATP synthase) to catalyze the formation of ATP. In this study, proteins involved in providing cellular energy were up-regulated to provide energy for mycorrhizal plants to resist salt stress.

### 3.6. Photosynthesis-Related Proteins

AMF and plant symbiosis can promote plant growth by increasing the photosynthesis of plants under salt stress, thereby increasing the ability of plants to resist the salt stress [48,49,50,51,52,53]. In this study, a photosystem II D1 precursor processing PSB27 was found to have upregulated expression. Photosystem II D1 protein is the core protein of photosystem II, which is synthesized in light and is injured by light or other adverse factors. Repairing D1 protein damage repairs PS II, maintaining its dynamic balance through continuous turnover [54,55]. In addition, the reactive center proteins, D1 and D2, are the binding sites of the auxiliary factors for all redox activities, which are necessary for PSII electron transport [54,56]. Thus, the photosystem II D1 protein plays an important role in the maintenance of the stability of the PSII reaction center. The previously mentioned thioredoxin (TRX), which is electronically reduced by the ferredoxin/thioredoxin system (Fd/TRX) in the chloroplasts of plants, is involved in the electron transport of plant photosynthesis. Guo [57] and other studies found that GRXs gene-silenced plants resulted in a significant decrease in ETR of photosystem II, and a significant increase in NPQ in varying degrees. Meanwhile, GRXs gene silencing results in a significant decrease in the maximum quantum efficiency (Fv/Fm) and the actual electronic yield (Ф PSII) under high temperature stress. This is consistent with the results of the chlorophyll fluorescence parameters in the early stage of this experiment [58], in which Fv/Fm, PSII, NPQ, and ETR in the leaves of *E. angustifolia* inoculated with AMF were higher than those of non-mycorrhizal plants under salt stress. In this study, TRX and GRX were up-regulated at the protein level, and Fv/Fm, PSII, NPQ, and ETR of mycorrhizal plants were significantly increased at the apparent level, thus alleviating the damage of salt stress to plants.

### 3.7. Network Interaction Predictions Based on Differential Expression

The protein–protein interaction information of the studied proteins was retrieved from the IntAct molecular interaction database (http://www.ebi.ac.uk/intact/) by their gene symbols or STRING software (http://string-db.org/). These interactions included the direct physical proteins and the indirect proteins correlated with indirect functions as shown in Figure 10. The results showed that ubiquitin-60S ribosomal was the most correlated protein, directly or indirectly, with connections to proteins, such as 60S acidic ribosomal, elongation factor mitochondrial, ubiquitin-activating enzyme E1, and E3 ubiquitin-ligase. Additionally, most of the target proteins were associated with ubiquitin-60S ribosomal and were in the protein synthesis, folding, and degradation pathway. Thus, this pathway might play an important role in stress tolerance. Ubiquitin-60S ribosomal interacts with various peroxidases, thereby participating in *E. angustifolia* protein synthesis, folding, and degradation biosynthetic processes, and modulates stress responses. These findings showed that the salt stress response is a multi-factor process involving many protein interactions.

The hypothesized mechanism of the improved salt tolerance of the mycorrhizal plant was revealed from the proteome. 

In the current study, based on proteomic data analysis, it is suggested that AMF can improve the salt tolerance of *E. angustifolia* seedlings (Figure 11).
AMF accelerates the secondary metabolism of plants, mainly phenylpropanoid metabolism (shikimate O-hydroxycinnamoyltransferase, cinnamyl alcohol dehydrogenase, and caffeoyl-CoA O-methyltransferase), reducing salt damage to plants.AMF enhances the signal transduction of the second messenger Ca^2+^ (G protein, phospholipase C, plasma membrane Ca^2+^ transporter ATPase (PMCA), calcium binding (CML), calcium-dependent kinases (CDPKs), and calmodulin (CaM), increasing the speed of sensing and transmitting of stress signals, allowing plants to follow up.Among a variety of antioxidant pathways (TRX/GRX, GST/GPX, POD), AMF enhances the antioxidant capacity of plants by increasing ROS clearance.AMF promotes protein biosynthesis, speeding up protein folding, and inhibiting protein degradation (ubiquitin-60S ribosomal, 60S acidic ribosomal, elongation factor mitochondrial, ubiquitin activating enzyme E1, and E3 ubiquitin- ligase).In the chloroplast, AMF maintains the PSII reaction centre conformation stability and speeds up photosynthetic electron transport (TRX/GRX, photosystem II D1 precursor processing PSB27- chloroplastic); in mitochondria, AMF enhances the synthesis of ATP (NADH dehydrogenase, cytochrome C oxidase, iron sulfur protein NADH dehydrogenase, and ATP synthase), providing sufficient energy for cellular activities.


## 4. Materials and Methods 

### 4.1. Experimental Materials and Salinity Treatments

*E. angustifolia* seeds were provided by Heilongjiang Jinxiudadi Biological Engineering Co. Ltd. (Haerbin, China). AM fungus *Rhizophagus irregularis* (RI) was propagated and preserved by Heilongjiang Provincial Key Laboratory of Ecological Restoration. The mycorrhizal inoculum containing approximately 25–30 AM propagules/g consisted of soil, spores, mycelia, and infected root fragments. The soil was collected from the Forest Botanical Garden of Heilongjiang Province, sieved (5 mm), and mixed with vermiculite (3:1, soil:vermiculite, *v*/*v*). The soil medium was pre-autoclaved at 121 °C for 2 h.

There were four different treatments as follows: *E. angustifolia* inoculated *Rhizophagus irregularis* without salt stress, *E. angustifolia* inoculated *Rhizophagus irregularis* under salt stress (300 mmol/L NaCl), *E. angustifolia* non-inoculated *Rhizophagus irregularis* without salt stress, and *E. angustifolia* non-inoculated *Rhizophagus irregularis* under salt stress (300 mmol/L NaCl). Each treatment had six replicates. The inoculated dosage of mycorrhizal inoculum per pot was 10 g. The same amount of inactive mycorrhizal inoculum (121 °C, 2 h) was used in non-inoculated treatments. The 300 mmol/L NaCl was added into the pots after 4 months of being inoculated with mycorrhizal inoculum as described [47]. Seedlings continued to be cultivated for 30 days. The experiment was carried out under outdoor natural conditions. 

Three seedlings were randomly selected from each pot and 6–7 round leaves were removed from each seedling. Leaves from two pots of the same treatment were combined as one sample. There were three biological repeats per treatment. The proteomic, AMF colonization, and growth parameter of the samples was detected.

### 4.2. Measurement of AMF Colonization and Growth Parameter

The AMF colonization rate of *E. angustifolia* was determined by the acid fuchsin staining method [59]. The height and diameter of *E. angustifolia* were measured by a vernier caliper. The root area and root length of *E. angustifolia* were measured by a root scanner.

### 4.3. Measurement of Soluble Sugar Content, Soluble Protein Content in the Leaves of E. angustifolia under Salt Stress

To assess the degree of stress, the contents of sugar and soluble protein were detected using the anthrone colorimetric method and coomassie brilliant blue G-250 method [60,61]. Samples were taken during the same growth period and from the same leaf positions.

### 4.4. Extraction and Quantification of Proteins

The leaves of all samples were frozen in liquid nitrogen and ground with a pestle and mortar. TCA/acetone (1:9) was added to the powder (1:5, *v*/*v*) and mixed by vortex. The mixture was placed at –20 °C for 4 h, and centrifuged at 6000× *g* for 40 min at 4 °C. The supernatant was discarded. The pre-cooling acetone was added into the pellet and washed three times. The precipitation was air dried. SDT buffer (1:30, *v*/*v*) was added to 20–30 mg of powder, mixed, and boiled for 5 min. The supernatant was filtered with 0.22 µm filters. The filtrate was quantified with the BCA Protein Assay Kit (P0012, Beyotime, Shanghai, China). 

### 4.5. FASP Digestion

Proteins of each sample (200 μg) were mixed with 200 μL UA buffer, and the mixture was filtered by a ultrafiltration centrifugal tube (10 kD) at 14,000× *g* for 15 min, and the pellet was re-suspended and filtered. The 100 μL iodoacetamide (100 mM IAA in UA buffer) was added into each sample and incubated for 30 min in darkness. The filters were washed with 100 μL UA buffer three times and subsequently with 100 μL 25 mM NH_4_HCO_3_ buffer twice. The protein suspensions were digested with 4 μg trypsin in 40 μL 25 mM NH_4_HCO_3_ buffer overnight at 37 °C. The collection of the resulting peptides with a new collector and peptides were desalted on C18 Cartridges, freeze-dried, and reconstituted in 40 µL of 0.1% (*v*/*v*) formic acid. The peptide content was estimated by UV light spectral density at 280 nm.

### 4.6. LC-MS/MS Analysis

Proteins were separated by using an Easy nLC HPLC liquid phase system with an increasing flow rate. The chromatographic column was balanced by 95% A solution (0.1% formic acid aqueous solution). The sample was injected onto the C18 column (Thermo Scientific Acclaim PepMap100, 100 μm × 2 cm, nanoViper C18) by an automatic sample injector and separated by C18-a2 analytical column (Thermo scientific EASY column, 10 cm, ID75 μm, 3 μm, C18-A) at a flow rate of 300 mL/min. Solution B (0.1% formic acid acetonitrile aqueous solution) was then used for gradient elution.

The samples separated by chromatography were analyzed with a Q-Exactive mass spectrometer. The analysis time was 120 min, the detection method was the positive-ion mode, the parent-ion scanning range was 300–1800 *m*/*z*, the resolution of a first-order mass spectrometer was 70,000 at 200 *m*/*z*, the AGC target was 3e6, the first level maximum IT was 10 ms, the number of scan ranges was 1, and the dynamic exclusion was 40.0 s.

### 4.7. Database Search and Protein Quantification

The database used was P16440_Unigene.fasta.transdecoder_73797_20161212.fasta (Sequence 73797, self-building). Maxquant software 1.3.0.5 [62] was used to analyze the protein qualitatively and quantitatively in the original raw file. The maxquant software parameter table is shown in Table 3. 

For the proteins identified by mass spectrometry in the original data, differentially expressed proteins and differentially expressed proteins were screened by the screening criteria of Ratio > +/−2 and *p* value < 0.05. The quantified protein sequence information was extracted in batches from the UniProtKB database (version number: 201701).

### 4.8. Protein GO Functional Annotation and KEGG Pathway Annotation

Blast 2 GO was used to annotate the functions of the targeted proteins [63]. KASS software was used for pathway analysis. The target protein sequences were classified by KO compared to the KEGG GENES database, and then the pathway information involved in the target protein sequence was automatically acquired according to the KO classification.

### 4.9. Protein—Protein Interact Network (PPI)

The protein—protein interaction information of the studied proteins was retrieved from the IntAct molecular interaction database (http://www.ebi.ac.uk/intact/) by their gene symbols or STRING software (http://string-db.org/). The results were downloaded in the XGMML format and imported into Cytoscape5 software (http://www.cytoscape.org/, version 3.2.1) to visualize and further analyze the functional protein-protein interaction networks. Furthermore, the degree of each protein was calculated to evaluate the importance of the protein in the PPI network.

## 5. Conclusions

*E. angustifolia* seedlings’ growth was significantly inhibited by salt stress, and growth was improved in mycorrhizal symbionts. Furthermore, mycorrhizal *E. angustifolia* had a higher leaf soluble sugar and soluble protein content than that of the non-mycorrhizal seedlings during salt stress. Additionally, it was found that AMF inoculated *E. angustifolia* seedlings increased secondary metabolism, enhanced Ca^2+^ signal transduction and ROS scavenging capacity, promoted protein biosynthesis, accelerated protein folding, and inhibited protein degradation compared with non-inoculated plants under salt stress. In addition, AMF maintained the conformation stability of the PS II reaction center, accelerated the photosynthetic electron transfer, enhanced ATP synthesis, and provided sufficient energy for cell activity. Overall, these findings show that AMF played an important role in easing salt stress in plants and contributed to saline alkali soil remediation.

## Figures and Tables

**Figure 1 ijms-20-00788-f001:**
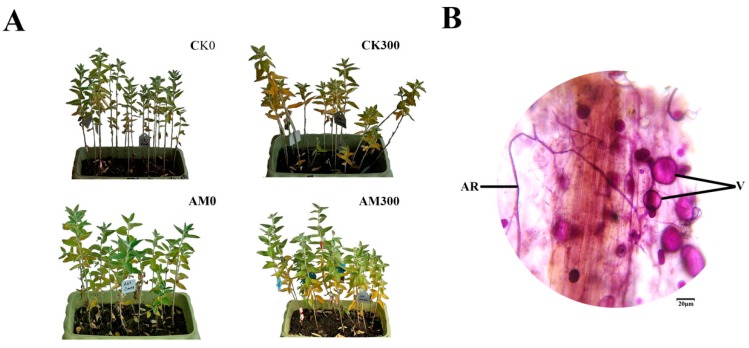
Growth of *E. angustifolia* inoculated AMF (arbuscular mycorrhizal fungi) under salt stress and a representative image of AMF colonization. Note: (**A**) represents the growth contrast in mycorrhizal and non-mycorrhizal *E. angustifolia* after salt stress. (**B**) represents a photomicrograph of the structural colonization of AMF in the root of *R. irregularis*. AM, mycorrhizal; CK, non-mycorrhizal; 0 mmol/L, without salt stress; 300 mmol/L, during salt stress; AR: Arbuscule; V: Vesicles. Scar bar: 20 μm.

**Figure 2 ijms-20-00788-f002:**
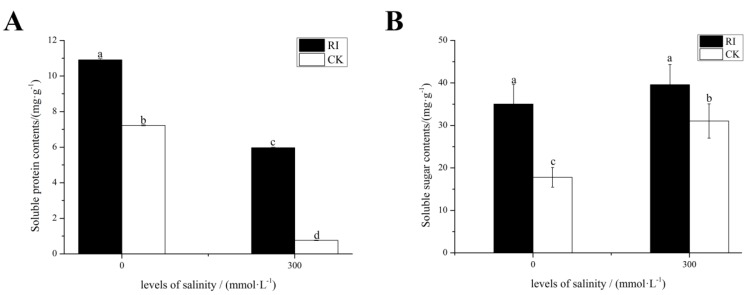
Effects of RI and CK on the soluble sugar content and soluble protein content in the leaves of *E. angustifolia* under salt stress. Note: (**A**) soluble protein, (**B**) soluble sugar. RI, mycorrhizal; CK, non-mycorrhizal; 0 mmol/L, without salt stress; 300 mmol/L, during salt stress. Columns represent the means for three replicates (*n* = 3). Error bars show the standard error. Columns with different letters indicate significant differences between the treatments at *p* < 0.05.

**Figure 3 ijms-20-00788-f003:**
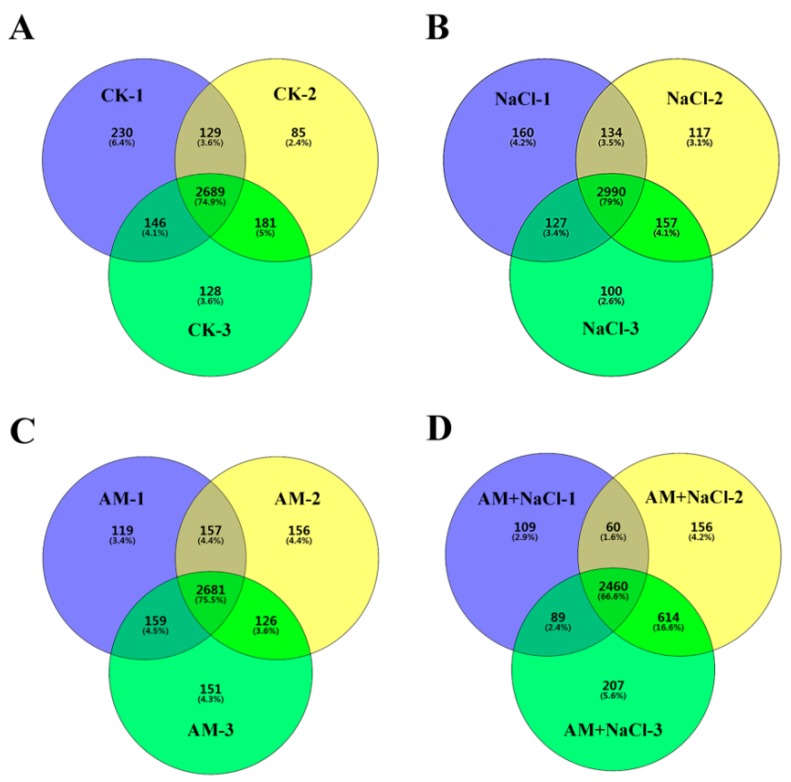
Statistics of the number of proteins identified in each treatment group. Note: (**A**) the number of proteins identified in the three replicates of CK group; (**B**) the number of proteins identified in the three replicates of NaCl group; (**C**) the number of proteins identified in the three replicates of AM group; (**D**) the number of proteins identified in the three replicates of AM + NaCl group.

**Figure 4 ijms-20-00788-f004:**
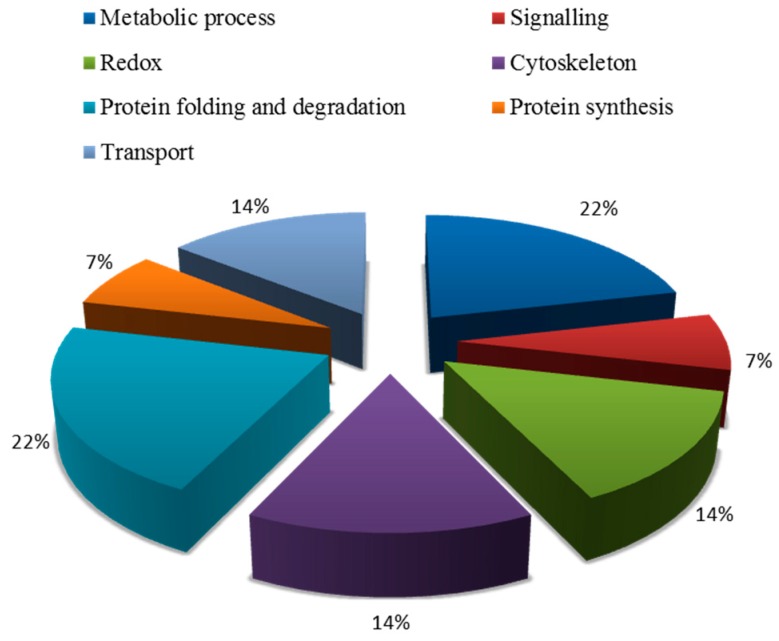
Biological functional classification of salt-tolerant proteins induced by symbiosis.

**Figure 5 ijms-20-00788-f005:**
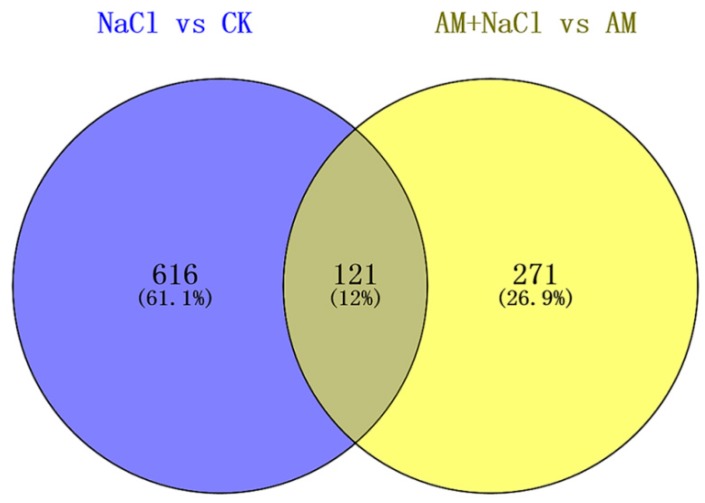
Venn diagram of the protein differential expression between the NaCl/CK group and AM + NaCl/AM group.

**Figure 6 ijms-20-00788-f006:**
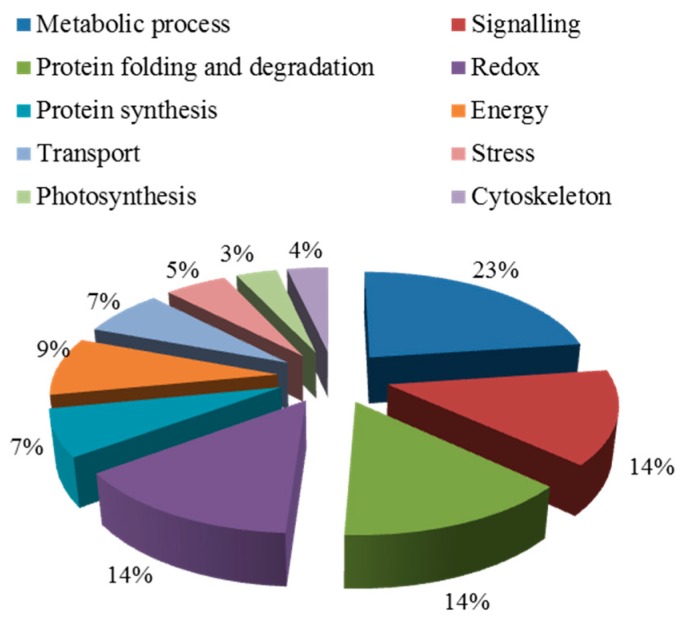
Biological functional classification of salt-induced mycorrhizal protein.

**Figure 7 ijms-20-00788-f007:**
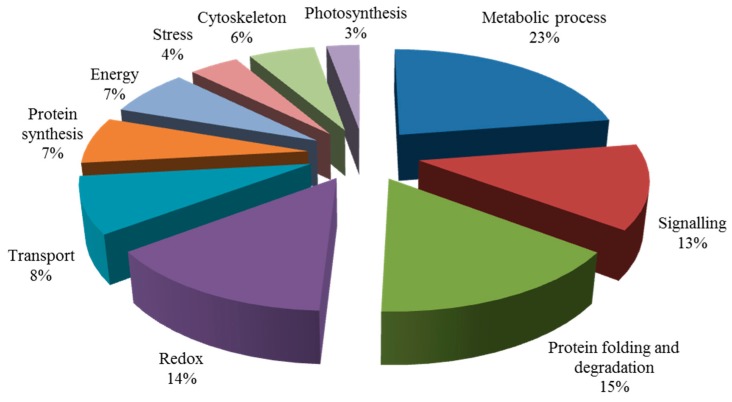
Biological functional classification of differential proteins.

**Figure 8 ijms-20-00788-f008:**
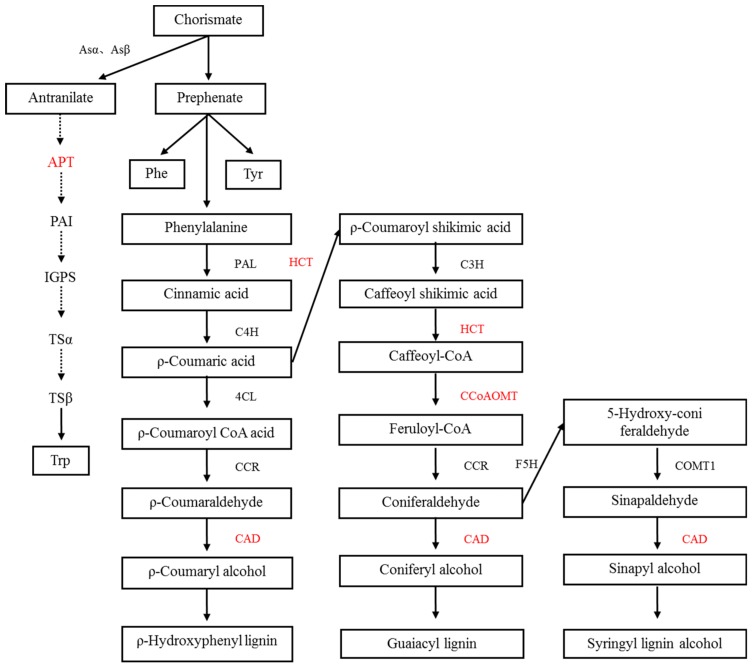
Secondary metabolism of the alleviation of salt stress in mycorrhizal *E. angustifolia* seedlings. Note: red color items represents proteins related to secondary metabolism in this study.CM: chorismate mutase; APT: anthranilate phosphoribosyltransferase; HCT: shikimate O-hydroxycinnamoyltransferase; CAD: cinnamyl alcohol dehydrogenase; CCoAOMT: caffeoyl-CoA O-methyltransferase.

**Figure 9 ijms-20-00788-f009:**
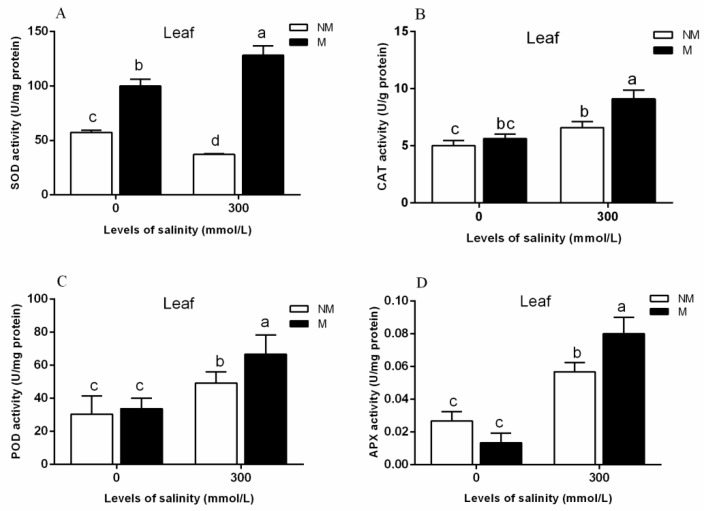
Effects of AMF inoculation on the superoxide dismutase (SOD) (**A**), catalase (CAT) (**B**), peroxidase (POD) (**C**), and ascorbate peroxidase (APX) (**D**) activities in the leaves during different salt conditions. M, mycorrhizal; NM, non-mycorrhizal; 0 mmol/L, without salt stress; 300 mmol/L, during salt stress. Columns represent the means for three plants (*n* = 3). Error bars show the standard error. Columns with different letters indicate significant differences between treatments at *p* < 0.05. Note: cited from “Arbuscular Mycorrhizal Symbiosis Modulates Antioxidant Response and Ion Distribution in Salt-Stressed *Elaeagnus angustifolia* Seedlings” [47].

**Figure 10 ijms-20-00788-f010:**
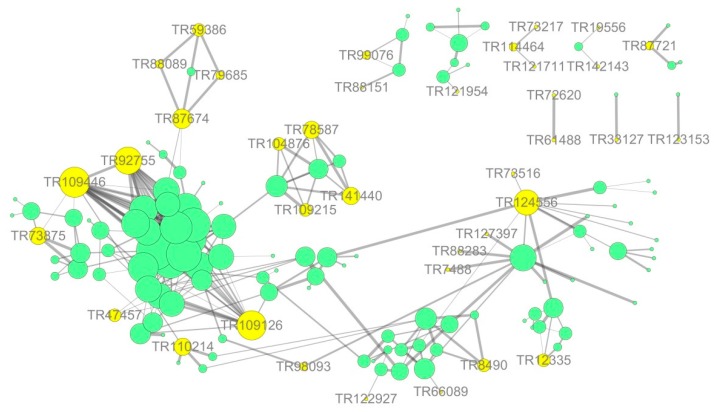
Protein—interaction network interactions for differentially expressed proteins. Note: shows the integrated network for all the differentially expressed proteins; yellow circle represents salt tolerance-related proteins induced by symbiosis and salt-induced mycorrhizal protein in this study; green circle represents other differentially expressed proteins in this study, respectively. The sizes represent the abundance of differentially expressed proteins.

**Figure 11 ijms-20-00788-f011:**
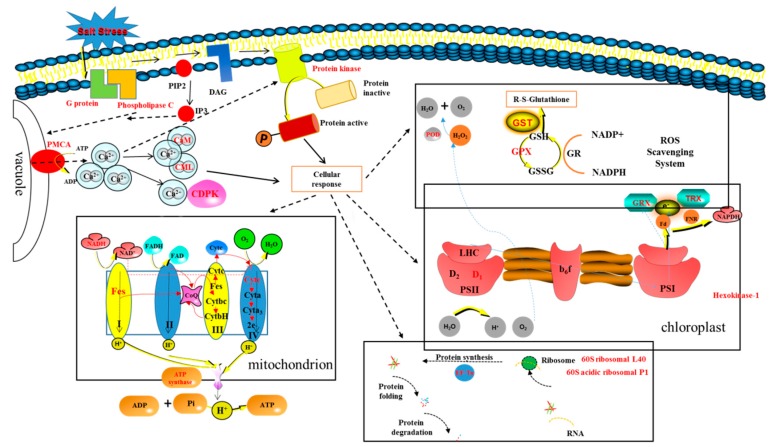
Mechanisms of mycorrhizal *E. angustifolia* seedlings to alleviate salt stress. Note: Protein folding and degradation: peptidyl-prolyl cis-trans isomerase FKBP12; FKBP-type peptidyl-prolyl cis-trans isomerase 5 isoform 1; peptidyl-prolyl cis-trans isomerase CYP18-1; peptidyl-prolyl cis-trans isomerase FKBP62; prefoldin subunit 1; prefoldin subunit 2; Heat shock 70 kDa partial; small heat shock protein 17.3 kDa; E3 ubiquitin ligase (UPL3); ubiquitin-like 1-activating enzyme (E1 B). PMCA: calcium-transporting ATPase; CaM: calmodulin; CML: calcium-binding CML20; CDPK: calcium-dependent kinase; POD: peroxidase; GST: glutathione S-transferase; GPX: glutathione peroxidase; GRX:glutaredoxin; TRX:thioredoxin; D1:photosystem II D1 precursor processing PSB27; EF-Tu: elongation factor mitochondrial; NADH: NADH dehydrogenase [ubiquinone] 1 beta subcomplex subunit 7; Fes: NADH dehydrogenase [ubiquinone] iron-sulfur mitochondrial; Cytc: cytochrome c oxidase subunit mitochondrial.

**Table 1 ijms-20-00788-t001:** Effects of RI and CK on the height, diameter, and roots of *E. angustifolia* under salt stress.

Level of Salinity/(mmol/L)	Different Treatment	Height/(cm)	Diameter/(mm)	Length/(cm)	Area/(cm^2^)
0	CK	45.50 ± 0.24c	5.65 ± 0.17b	985.73 ± 27.80b	146.04 ± 5.98c
	RI	49.07 ± 0.54a	6.54 ± 0.20a	1256.7 ± 22.52a	213.07 ± 13.04a
	Significance	**	**	**	**
300	CK	39.57 ± 0.26f	3.99 ± 0.14e	763.64 ± 23.34e	93.68 ± 6.27e
	RI	43.17 ± 0.21de	4.82 ± 0.11d	896.56 ± 42.36bcd	126.96 ± 8.03cd
	Significance	**	**	*	**

RI, mycorrhizal; CK, non-mycorrhizal; 0 mmol/L, without salt stress; 300 mmol/L, during salt stress. Data are means ± SD (standard deviation) of six replicates. The same letter within each column shows no significant differences among treatments (*p* < 0.05). Levels of significance: * *p* < 0.05, ** *p* < 0.01.

**Table 2 ijms-20-00788-t002:** Differentially expressed proteins between treatments.

Treatments	Number of Differential Proteins
NaCl vs CK variation analysis	402 a + 335 b
AM vs CK variation analysis	35 a + 152 b
AM + NaCl vs AM variation analysis	166 a + 226 b
AM + NaCl vs NaCl variation analysis	62 a + 189 b

a: The number of proteins was the satisfied condition (ratio > ±2 and *p* value < 0.05); b: The number of proteins was only detected at CK or treatments (NaCl, AM, AM + NaCl).

**Table 3 ijms-20-00788-t003:** Maxquant software parameter table.

Item	Value
Main search ppm	6
Missed cleavage	2
MS/MS tolerance ppm	20
De-Isotopic	True
Enzyme	Trypsin
Database	P16440_Unigene.fasta.transdecoder_73797_20161212.fasta
Fixed modification	Carbamidomethyl (C)
Variable modification	Oxidation(M), Acetyl (Protein N-term)
Decoy database pattern	reverse
LFQ	True
LFQ min. ratio count	1
Match between runs	2min
Peptide FDR	0.01
Protein FDR	0.01

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
