# Peer review of "Proteomics Analysis of E. angustifolia Seedlings Inoculated with Arbuscular Mycorrhizal Fungi under Salt Stress"

_ijms, 2019, doi:10.3390/ijms20030788_

Round 1
Reviewer 1 Report
The manuscript, “Proteomics analysis of E. angustifolia seedlings inoculated with arbuscular mycorrhizal fungi under salt stress” is a nice effort to reveal the mechanism of salinity stress alleviation by arbuscular mycorrhizal fungi (AMF). After going through the manuscript fully, the main suggestion will be to represent the nmanuscript in a better way. The manuscripts most of the times loses its continuity. For example, the Methods and Results sections are not well coordinated. Please take care of it. Following are a few suggestions where the manuscript needs improvements.
Comments and Suggestions:
1. There are several places where the sentence “the mycorrhizal seedlings grew better than the non-mycorrhizal seedlings” has been written. Modify accordingly.
2. In the line no. 117, the authors have mentioned “As shown in Table 2, a total of 186 differentially expressed proteins were identified……….…………..17 proteins were identified as symbiotic salt tolerance related proteins after reference to many academic document”. No references have been provided. Which documents are the authors talking about?
3. The authors have mentioned about biological functions. Do they get them from Gene Ontology (GO) analysis? If so, mention it clearly. Also for GO analysis we get 3 components such as a. Biological Process, b. Cellular Component and c. Molecular Function. Which one or all of them have they used?
4. In the line no. 143, “AM+NaCl vs AM groups by Wayne diagram”, do the authors mean Venn diagram?
5. In the line no. 489-490, modify the sentences “The parameter settings are shown in Table 5. The parameter settings are shown in the identified proteins, Ratio>+/-2 and P value<0.05 was used as screening criteria to determine whether or not these proteins were differentially expressed.”
6. The protein-protein interaction network could have been mentioned in the result section and the respective methods in the materials and methods section. Also please change the color of the nodes or put a boundary color to be clearly visible.
7. Also the network could have been presented in more detail. Number of nodes and edges, degree of the nodes (may have connected to that the ubiquitin-60S is the most correlated node), etc.
8. In the caption of Figure-10, the authors have mentioned that the nodes with yellow colors are the differentially expressed proteins from their study and the green color nodes are from??? Mention in detail.
Author Response
Dear Reviewer,
Thank you very much for your valuable comments on our manuscript. Please find our response cover letter in the attachment.

Reviewer 2 Report
The authors have conducted a study comparing proteomic changes in E.angustifolia under salt stress and effect of arbuscular mycorrhizal fungi on the plant with and without salt stress. The study is interesting but can be improved.
Introduction: I feel the introduction is poorly written, it doesn't create enough emphasis on the importance of the study. It is creating enough interest to the reader, it sounds very generic, I suggest rewriting the whole section.
Do AMF colonizes E.angustifolia in nature?
Line 41:
"Plants are subjected to salt stress in the presence of high salinity in the presence of soil" - what do you mean by this? Rephrase
Methods:
Methods are not described adequately enough for repetition and experimental design must be improved.
- How many plants were used per pot? Did you consider each pot as a single replicate or one plant as replicate? How many pots were used per each treatment?
- " 300mmol/L" - how much solution was used per pot?
- Explain in more detail " it took 6 days to reach the desired levels"? Have started to water with low concentration solutions first? Why do you need to follow this approach?
- Have you watered the plants for the next 30 days after treating with salt?
- Is it possible that salt is washed off during watering for the next 30 days?
- Why did you choose only 300 mM concentration? Why only one concentration?
- Did you run the protein on the gel? Do you have comparison pictures?
- Line 453: "For each plant, the leaves of amount samples were frozen " What is amount samples?
- Line 454: " 5 mines volume" ??
- Have pooled the leaves from different plants in the same treatment? or leaves are from different plants?
Results:
Fig 1A: It appears the pictures have different brightness settings, top 2 appears dull compared to the bottom panel, use pictures with same light conditions.
Fig 2A: Y-axis legend typo, should be "soluble".
Provide abundance data for all the differentially expressed proteins and fold changes in the supplementary section in a table.
Overall, the inclusion of at least 2 additional salt concentrations and 2 more time periods for collection samples would be more interesting and will further confirm the results.
Author Response
Thank you very much for your valuable comments. Please find the response cover letter in the attachment.

Round 2
Reviewer 2 Report
The manuscript "Proteomics analysis of E. angustifolia seedlings inoculated with arbuscular mycorrhizal fungi under salt stress" is interesting however it lacks more treatment groups or time points for comparison. Proteomic changes cannot be accurately concluded based on one single treatment i.e 300mM NaCl. Experimental methods also need improvements. Authors used too many plants in the single pot and they accept that there is a possibility of leaching of salt due to watering for 30 days. If the salt is leached out, it can't be concluded that all the protein changes observed are the direct consequence of 300mM salt conc. Authors should have included the data from their other concentrations that they have tested prior to fixing on 300mM.
Author Response
Thank you very much for your positive and constructive comments on our experimental design. we carried out the pre-experiment of salt stress of 100mmol/L, 200mmol/L, 300mmol/L and 400mmol/L. The results showed that there were significant differences in the apparent traits of E.angustifolia under 300mmol/L salt stress, and all E.angustifolia seedlings would die under 400mmol/L salt stress. Therefore, we chose 300 mmol/L concentration only considering the apparent difference of E.angustifolia for this experiment. The pot was used (the total volume of the pot is about 0.006m3) is a bottom-closed pot in this experiment. Under the condition of bottom-closed, it will not lead to the loss of salt content, which can ensure the consistency of salt stress conditions of all treatments in the experiment. In the future, we have also listened to your opinions modestly, the experiment will continue to be carried out in depth.